# Technical Note: colab_zirc_dims: a Google-Colab-compatible toolset for automated and semi-automated measurement of mineral grains in LA-ICP-MS images using deep learning models

Michael C. Sitar[1], Ryan J. Leary[2]

[1]Department of Geosciences, Colorado State University, Fort Collins, CO, 80523-1482, USA
[2]Department of Earth and Environmental Science, New Mexico Institute of Mining and Technology, Socorro, NM, 87801, USA

*Correspondence to*: Michael C. Sitar (mcsitar@colostate.edu)

**Abstract**

Collecting grain measurements for large detrital zircon age datasets is time-consuming, but a growing number of studies suggest such data are essential to understanding complex roles of grain size and morphology in grain transport and as indicators for grain provenance. We developed the colab_zirc_dims Python package to automate deep-learning-based segmentation and measurement of mineral grains from scaled images captured during laser ablation at facilities that use Chromium targeting software. The colab_zirc_dims package is implemented in a collection of highly interactive Jupyter notebooks that can be run either on a local computer or installation-free via Google Colab. These notebooks also provide additional functionalities for dataset preparation and for semi-automated grain segmentation and measurement using a simple graphical user interface. Our automated grain measurement algorithm approaches human measurement accuracy when applied to a manually measured n = 5,004 detrital zircon dataset. Errors and uncertainty related to variable grain exposure necessitate semi-automated measurement for production of publication-quality measurements, but we estimate that our semi-automated grain segmentation workflow will enable users to collect grain measurement datasets for large (n ≥ 5,000), applicable image datasets in under a day of work. We hope that the colab_zirc_dims toolset allows more researchers to augment their detrital geochronology datasets with grain measurements.

## 1 Introduction

Despite an increasing number of studies on the subject, the degree to which detrital geochronology datasets are affected by sample and mineral grain size remains unresolved. Several detrital zircon studies have documented substantial grain size-dependent mineral fractionation leading to biased detrital age spectra and erroneous provenance interpretations (e.g., Lawrence et al., 2011; Ibañez-Mejia et al., 2018; Augustsson et al., 2018; Cantine et al., 2021). Conversely, several other studies have identified provenance-dependent grain size relationships in detrital samples with little evidence of age spectra biasing by selective transport processes (e.g., Muhlbauer et al., 2017; Leary et al., 2020a, 2022). Because the number of studies

characterizing grain size of detrital zircon datasets remains relatively small, especially compared to the number of studies employing detrital zircon geochronology, we likely lack the necessary volume and diversity of datasets to understand under which specific circumstances zircon transport processes will bias age spectra and interpreted provenance (Leary et al., 2022). A principal challenge in collecting such data has been that few automated approaches have been published (e.g. Scharf et al., 2022), and the time required to manually collect grain dimensions from large detrital datasets is a substantial barrier to

widespread application of these methods (e.g. Leary et al., 2020a).

        Zircon grains can be measured manually using analogue methods prior to LA-ICP-MS, but doing so is prohibitively time consuming. Grains may also be imaged, characterized, and measured via scanning electron microscope before or after analysis, but this too incurs time and instrumentation costs that increase with sample size, and such analyses are not standard at most labs. Many LA-ICP-MS facilities using Teledyne-Photon machines laser ablation systems with proprietary Chromium

(Teledyne Photon Machines, 2020) targeting software save reflected light images of samples during analysis with scaling and shot location metadata files and provide these files to facility users. Images from these facilities may be full-sample mosaics captured prior to analyses or single, grain-centred per-shot images captured during ablation. The former are provided by the University of Arizona LaserChron Center (ALC) and the latter by the University of California, Santa Barbara (UCSB) Petrochronology Center. Many researchers who have not otherwise imaged their large-n detrital mineral datasets do have

access to these files, and these can be used to locate and manually measure detrital mineral grains using the offline version of the Chromium targeting software (Leary et al., 2020a).

        Three limitations to manual grain measurement in Chromium (Leary et al., 2020a) are a) grains may be partially exposed or over-polished at the surfaces of epoxy mounts, so measurements are minimum, rather than true dimensions, b) this method is extremely time consuming, and c) this method can only produce one-dimensional (i.e., length) measurements. The

first problem is inherent to reflected light images, but the latter two can be mitigated and solved, respectively, via automated two-dimensional grain-image segmentation and measurement of segmentation results. Deep learning methods, wherein training-optimizable models are used to algorithmically extract information from data (e.g., images) with minimal pre-processing (Alzubaidi et al., 2021), are at the cutting edge of accuracy in image segmentation and so allow grain image segmentation to be automated to a greater degree than other methods (e.g., thresholding).

We developed the colab_zirc_dims Python package, which contains code to automatically segment and measure mineral grains from Chromium-scaled LA-ICP-MS reflected light images using deep learning instance segmentation (i.e., where grains are treated as separate objects and distinguished from one another) models. Such models are computationally expensive to run and can be quite slow without a good, code-compatible graphics processing unit (GPU). In order to maximize its accessibility, we implemented our code in Jupyter notebooks (i.e., Kluyver et al., 2016) that can be run either offline or

online and installation-free using Google Colab (Sitar, 2022). Google Colab is a free service that allows users to run Jupyter notebooks on cloud-based virtual machines with variably high-end GPUs from the NVIDIA Tesla series (i.e., K80, T4, P100, and V100) that are allocated based on availability. Because its user interface is notebook-based, colab_zirc_dims is not a per-se application but a set of simplified, highly interactive scripts that rely on a backend of code in the colab_zirc_dims package.

Deep-learning-based techniques are increasingly applied to geologic image segmentation tasks such as fission track counting (Nachtergaele and De Grave, 2021), cobble measurement (Soloy et al., 2020), and photomicrograph grain segmentation (e.g., Bukharev et al., 2018; Filippo et al, 2021; Jiang et al., 2020; Latif et al., 2022). We expect such techniques to continue to proliferate in the future, but the colab_zirc_dims package and processing notebooks represent, to the best of our knowledge, the first deep-learning-based approach to per-grain detrital mineral separate measurement.

## 2 Established image segmentation techniques and related software

Automated segmentation of mineral grains in LA-ICP-MS images can be achieved with some success using relatively simple image segmentation techniques such as k-means clustering, edge detection, and intensity thresholding. Otsu's thresholding method (i.e., Otsu, 1979), wherein image pixels are automatically segmented into background and foreground classes via maximization of inter-class intensity variance, is particularly well-suited for reflected light images because mineral grains appear as a bright phase against an epoxy background (Fig. 1). Although grain segmentations produced through Otsu thresholding are often accurate, they tend to split single fractured grains into multiple sub-grains (Figs. 1c, A1) and can be wildly inaccurate where image artefacts affecting pixel intensity (e.g., anomalous bright spots; Fig. A1) are present. These problems are common to automated segmentation techniques, and edge detection methods additionally contend with mis-segmentations along artefactitious edges where sub-image boundaries appear within larger, otherwise uniform mosaic images (e.g., Fig. A1). Because deep learning models can be optimized through training to ignore image artefacts and intra-grain fractures, they are likely the best available tool for achieving fully automated mineral grain segmentations with near-human accuracy.

Some existing software applications enable measurement of mineral grains in images with varying degrees of automation. The offline version of the Chromium LA-ICP-MS targeting application supports loading and viewing of scaled alignment images and shot locations; users can manually measure the axial dimensions of analysed grains using a ruler-like "measure" tool (Leary et al., 2020a; Teledyne Photon Machines, 2020). The ZirconSpotFinder module of the MATLAB-based AgeCalcML application likewise supports loading and viewing of Chromium-scaled LA-ICP-MS alignment images, but also implements semi-automated grain segmentation using user-selected thresholds, filtering of segmented grains by surface area, and export of area-filtered shot lists (Sundell et al., 2020). AnalyZr, a new application designed specifically for measurement of zircon grains in images, combines Otsu thresholding with a novel boundary separation algorithm to automatically segment grains and allows users to edit the resulting segmentations before exporting automatically-generated, grain-specific dimensional analyses (Scharf et al., 2022). Analytical spot identification and localization in AnalyZr is done manually through an interface that also allows input of spot-specific comments and qualitative internal grain zoning descriptors that persist into the program's exports (Scharf et al, 2022). Because AnalyZr supports loading of grain image .png files from any source with manual capture of image scale, it can be used to extract more detailed per-grain information (e.g., unobscured grain dimensions from transmitted light images) than is obtainable using only reflected light images (Scharf et al., 2022). AnalyZr's manual spot

placement and scaling implementations and thresholding-based segmentation algorithm also, however, necessitate substantial human involvement in producing accurate grain segmentations and measurements. The colab_zirc_dims package and notebooks are likely better suited for rapid measurement of mineral grains in applicable (i.e., with Chromium-scaled images) large-n datasets due to their automated image loading, scaling, and generally accurate deep-learning-based automated segmentation capabilities.

**Figure 1. Visualizations of image thresholding segmentation using Otsu's method (Otsu, 1979) and its inherent problems in the context of reflected light detrital zircon grain images (top row), and of the colab_zirc_dims segmentation and grain measurement process (bottom row). (a) An original, unaltered LA-ICP-MS reflected light image. (b) A binary image resulting from segmentation of the original image into foreground (white) and background (black) classes using Otsu's method. (c) The original image with "background" masked out using the binary image. Red highlights indicate single grains that have been erroneously eroded, segmented into multiple grains along fractures, or both. (d) The results (bounding boxes, probability scores, and masks) of instance segmentation of the original image using a Mask RCNN model (M-ST-C), as displayed by the Detectron2 "visualizer" module. (e) The resulting colab_zirc_dims verification image, scaled in µm and displaying the identified central grain mask (yellow), mask centroid (green), minimum-area circumscribing rectangle (blue), and ellipse with the same $2^{nd}$ order moments as the grain mask along with its axes (red).**

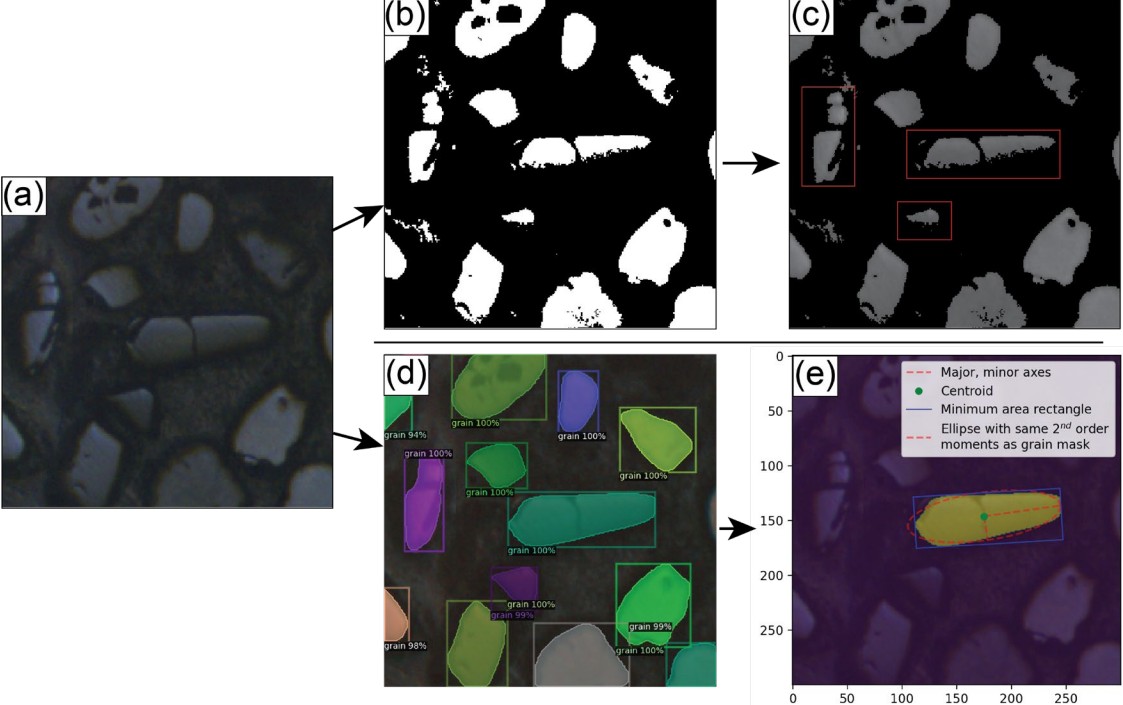

# 3 Methods

## 3.1 Dependencies

The colab_zirc_dims package was written in Python 3.8 and relies on some non-standard Python packages (Van Rossum, 2020). Pillow and Matplotlib are respectively used for image loading and to create and save verification segmentation images;

Matplotlib was additionally used to create figures for this manuscript (Umesh, 2012; J. D. Hunter, 2007). OpenCV (Bradski, 2000) is used to display images as well as to fit minimum-area circumscribing rectangles to masks (e.g., Figs. 1e, A1c). NumPy is used for array operations and conversions, and pandas is used in some contexts for data organization and export (Harris et al., 2020; McKinney, 2010). The "measure" module of Scikit-Image is used to produce unscaled dimensional analyses from segmented grain masks and to extract mask outlines for conversion into user-editable polygons (van der Walt et al., 2014). Interactivity in colab_zirc_dims processing notebooks is implemented using IPython (Pérez and Granger, 2007). Detectron2, which is a deep learning library that was developed by Facebook and is itself built on PyTorch, also developed by Facebook, was used for model construction and training and is used to deploy models within colab_zirc_dims processing notebooks (Paszke et al., 2019; Wu et al., 2019).

Local and online execution of the colab_zirc_dims notebooks rely, respectively, on Jupyter and Google Colab. We recognize that Jupyter-style notebooks are an unconventional platform for final deployment of scientific computing algorithms and that Google Colab in particular does have some significant disadvantages (e.g., runtimes will automatically disconnect if left idle for too long) versus deployment in a standalone, purpose-built local or web-based application. Nevertheless, we believe that Google Colab's benefits in this use-case outweigh its disadvantages, especially with regards to accessibility. The colab_zirc_dims notebooks can be run using otherwise-expensive GPUs by anyone with a Google account, regardless of their local hardware or prior Python experience. We also mitigate potential connection-related issues by implementing automatic saving to Google Drive during online automated and semi-automated grain-image processing: if a user's runtime disconnects, they can simply re-connect and resume work from the last sample processed before disconnection. The aforementioned timeout and connectivity problems will not affect the processing notebooks if they are run locally (i.e., Sitar, 2022, 'Advanced Local Installation Instructions'). Local notebook execution consequently remains an option for users who are equipped with suitable hardware and either chafe against the constraints of Google Colab or are otherwise unable to access Google services.

## 3.2 Training-validation dataset

We present "czd_large", a new training-validation dataset comprising 16,464 semi-automatically generated per-grain annotations in 1,558 LA-ICP-MS reflected light images of mineral grains (Table 1). Constituent images, which are sourced from both ALC and UCSB, were compiled via Chromium-metadata-informed (i.e., all images are non-overlapping in real-world space) random selection. ALC source mosaic images (Table 1) were captured during analyses of detrital zircon from the Eagle and Paradox basins, USA; dates and Chromium-derived manual grain measurements resulting from these analyses were published by Leary et al. (2020a). UCSB images (Table 1) were captured during unpublished analyses of detrital zircon from units in east-central Nevada, USA. Automatic per-grain instance segmentations were generated using a Mask-RCNN Resnet-101 model trained on a smaller, manually annotated dataset compiled from the same sources (Table B1; Sitar, 2022, 'Training Datasets'). These automatic segmentations were converted to the VGG image annotation format (Dutta and Zisserman, 2019) using a custom Python script, and annotations for every image were then manually reviewed and, where necessary, corrected or extended using the VIA Image Annotator (Dutta and Zisserman, 2019). Approximately 15% of the full dataset was split off

into a validation subset via sample-stratified random selection (Table 1). We provide granular information (e.g., image sizes and scales, training versus validation set image and annotation distributions, etc.) about the dataset and a link to download it in the 'Training Datasets' subdirectory of our project GitHub page (Sitar, 2022).

**Table 1. A summary of the "czd_large" dataset used to train the deep learning model presented in this manuscript for reflected light mineral grain segmentation. Please refer to Sitar (2022) for more in-depth information on the composition of the dataset.**

| Source facility | Training set images | Validation set images | Training set annotations | Validation set annotations |
|---|---|---|---|---|
| ALC | 1203 | 212 | 12923 | 2326 |
| UCSB | 121 | 22 | 1039 | 176 |
| Total | 1324 | 234 | 13962 | 2502 |
| | 1558 | | 16464 | |

Some training and validation images contain likely detrital apatite grains in addition to zircon, and we segmented all visible mineral grains into a single "grain" class to avoid harming our models' generalization abilities in the presence of varying image exposure and brightness levels. Models trained on "czd_large" are consequently likely applicable to segmentation of all reflected-light bright-phase minerals but are unable to distinguish these minerals from one another. Both automatically and manually generated annotations are conservative with regards to interpreting grain extent; we only segmented areas where grains are exposed above the epoxy surface except in cases where larger subsurface extents are incontrovertibly apparent.

**3.3 Deep learning models**

Using the "czd_large" dataset, we have trained several Detectron2-based instance segmentation models (i.e., configurations with trained weights) that can be applied in colab_zirc_dims processing notebooks. As of colab_zirc_dims v1.0.10, said models encompass several architectures and variations therein, including Mask-RCNN models with ResNet-FPN backbones, a Mask-RCNN model with a Swin-T backbone implemented using third-party code (Ye et al., 2021), and a Centermask2 model with a VovNetV2-99 backbone (Table B1). Given the rapid pace of progress in deep learning research and our own graceless yet continual progress in optimizing model hyperparameters for application in colab_zirc_dims, we expect that these models could be superseded by more performant models in the future. As such, we host our current models (i.e., configuration files and links to weights) and all explanatory information (i.e., training metrics, post-training evaluation metrics, and summary tables and diagrams) on a mutable 'Model Library' page within the project GitHub repository (Sitar, 2022). Users can refer to this page to learn more about the current selection of models, and to the linked Jupyter notebook files if they would like to train their own models using our training workflow. Models are loaded for application in local and Colab-based colab_zirc_dims processing notebooks through a dynamic selection and downloading interface. Our current default model is a Mask-RCNN model with a Swin-T-FPN backbone (Table B1), which was selected due to its apparent low propensity for producing

175 aberrantly over-interpretive segmentation masks (Sitar, 2022). This model is herein referred to a "M-ST-C" and was used to produce all measurements and segmentation images presented in the current study.

## 3.4 Dimensional analysis of mineral grains

The initial step in dimensional analysis of grains using colab_zirc_dims is standardized loading of grain images for segmentation such that differently formatted image-datasets can be processed using a single set of algorithms. Shot-centred
180 single images (e.g., from UCSB) can be passed to models for segmentation as-is, but segmentation of grains from mosaic image datasets (e.g., from ALC) is performed on scaled, shot-centred sub-images extracted from mosaics using shot coordinate metadata. Grain-centred images are segmented by a deep learning model, and the resulting segmentations (e.g., Figs. 2d, A1c) are passed to an algorithm that attempts to identify and return a "central" mask corresponding to the shot target grain LA-ICP-MS analysis (Fig. 2c). If no mask is found at the actual centre of the image, as may be the case in slightly misaligned images,
185 the algorithm searches radially outwards until either a mask is identified or the central ~10% of the image has been checked. To avoid erroneously returning significantly off-central (i.e., non-target) grains, the algorithm is considered to have "failed" if it cannot find a grain mask after this search, and null values are returned for the spot instead of shape parameters. If a central grain is found, its dimensions are analysed using functions from OpenCV (Bradski, 2000) and the scikit-image "measure" module (van der Walt et al., 2014). The resulting measurements and properties are, where applicable, scaled from pixels to μm
190 or μm$^2$ using a Chromium-metadata-derived scale factor.

   Successful grain-image processing by the colab_zirc_dims grain segmentation and measurement algorithm will return the following grain-mask properties: *area*, *convex area*, *eccentricity*, *equivalent diameter*, *perimeter*, *major axis length*, *minor axis length*, *circularity, long axis rectangular diameter, short axis rectangular diameter*, *best long axis length*, and *best short axis length*. Details on the derivation of all output grain-mask properties can be found on the 'Processing Outputs' section of
195 the colab_zirc_dims GitHub page (Sitar, 2022), but some properties merit further discussion. Circularity, for instance, is calculated from scikit-image-derived area and perimeter measurements using Eq. (1); this is a notably simpler and likely less robust calculation than would be required for grain roundness (i.e., Resentini et al., 2018).

**Equation 1:**

$$Circularity = \frac{4\pi * Area}{Perimeter^2}$$

   Major and minor axis lengths are calculated from the moments of the grain mask image and reported axes thus correspond to "the length of the… axis of the ellipse that has the same normalized second central moments as the region" (van der Walt et al., 2014). These axial measurements will consequently fit exactly to perfectly elliptical and circular grain masks but may be more approximate in the cases of rectangular and irregularly shaped grains (e.g., Fig. 1e). Rectangular diameter
205 measurements correspond to the long and short axes of the minimum area circumscribing rectangle (e.g., Fig. 1e) that can be fitted to a grain mask using the OpenCV minAreaRect function (Bradski, 2000). Minimum area rectangles will exactly fit to

rectangular grain masks, but in the case of more equant grains may be grossly misaligned from the grain axes that a human researcher would interpret. The two types of calculated axial measurement parameters each have drawbacks. To split the difference, we implement "best" long and short axis measurement fields. These fields return either moment-based or rectangle-based axial measurements depending on whether each grain mask's aspect ratio (i.e., moment-based long axis length divided by moment-based short axis length) is above or below an empirically chosen threshold of 1.8. Minimum-area bounding rectangles should trend towards coaxiality with moment-based axes with increasing aspect ratio, so rectangle-based measurements are returned for grain masks with higher aspect ratios and moment-based measurements are returned for those with lower aspect ratios.

## 4 Implementation

### 4.1 The colab_zirc_dims package

Code for loading and parsing Chromium alignment and shot list files, segmenting and measuring grains using deep learning models, and interacting with notebooks using widgets is contained within the colab_zirc_dims package. We have made this package available on the Python Package Index (Python Package Index - PyPI, 2022) for easy installation to local and virtual (i.e., Google Colab) machines. Some colab_zirc_dims modules (e.g., utilities for reading Chromium metadata files and basic segmentation functions) will work without Detectron2 and other bulky dependencies, but these must be installed for full functionality.

### 4.2 Dataset organization

Before using colab_zirc_dims notebooks to automatically or semi-automatically measure grains, users must set up a project folder containing their dataset (i.e., image and metadata files). If users plan to use colab_zirc_dims in Google Colab, they must then upload their project folder to Google Drive (Fig. 2A). Required formats for colab_zirc_dims project folders are simple but necessarily differ slightly between dataset types (e.g., ALC mosaics or UCSB per-shot images), and they are thoroughly documented in the processing notebook for each type of dataset. Once a project folder has been created and, optionally, uploaded to a user's Google Drive, they can proceed either directly to notebook-based processing in the case of per-shot image datasets or to an additional, likewise notebook-based dataset preparation step in the case of mosaic image datasets (Fig. 2A).

**Figure 2. A graphical summary of interfaces and workflow options available in colab_zirc_dims processing notebooks. Tasks that are handled automatically or semi-automatically by processing notebooks are shown in blue boxes. (a) A summary of possible dataset inputs which can be processed or made processable with the provided notebooks. (b) Summary of the workflow for preparing datasets for fully or semi-automated segmentation. (c) Summary of possible workflows for automated or semi-automated grain measurement and for exploratory visualization of the resulting measurements.**

**Per-sample mosaic datasets**
• 3-channel .bmp mosaic images with Chromium .Align files
+
• Chromium .scancsv shotlist files

**Image-per-shot datasets**
• 3-channel .jpg, .png, .bmp, or .tif shot-centred images
+
• Per-image Chromium .Align files
OR
• Known image scale(s)

**Other datasets**
• 3-channel .jpg, .png, .bmp, or .tif images
+
• Known image scale(s) (µm/pixel)

Align, scale, and/or select shot locations in Chromium Offline

Clip and save shot-centred images with unique file-names (e.g., using ImageJ)

**Dataset inputs** **(a)**

Create project directory, then:
• Upload project directory to Google Drive; run notebooks in Colab
OR
• Download, run notebooks locally (see GitHub installation directions)

**Dataset preparation** **(b)**

Create project directory

File names contain sample names? —No→ Organize files into subdirectories by sample

↓Yes

Place image files in a single subdirectory → .Align scaling files available?

*Mosaic_match* notebook
• Match .scancsv files to mosaics
• Correct .scancsv misalignment
• Define processed subimage size (*Max_grain_size* parameter)

Automatically generate *mosaic_info.csv* file

• Upload project directory to Google Drive; run notebooks in Colab
OR
• Download, run notebooks locally (see GitHub installation directions)

Yes / No → Add per-sample scales (µm/pixel) to *sample_info.csv* file

*Mosaic_grain_process* notebook

*Single_shot_image* notebook

**(c)**
**Dataset processing**

**Fully automated segmentation**
• Load dataset and select samples
• Automatically load models
• View trial segmentaion results
• Parameterize segmentation
• Run automated segmentation

**Outputs:**
• .csv files for each sample with measured per-grain shape parameters
• Mask polygon .json files for loading in GUI
• Verification images

**Semi-automated segmentation GUI**
• Load automatically generated grain mask polygons from current or prior session
OR
• Load semi-automatically generated polygons from a prior session
OR
• Automatically generate per-sample polygons on the go
THEN
• View, 'tag', and/or edit polygons
• Save polygons to project directory
• Generate and export measurements from polygons to project directory

**Exploratory Data Visualization UI**
• Load any colab_zirc_dims measurement results from project directory
• Filter dataset before plotting (e.g., include or exclude spot names based on string inputs)
• Parameterize, create, and view exploratory plots (bar-whisker, histogram, and scatter) of filtered dataset

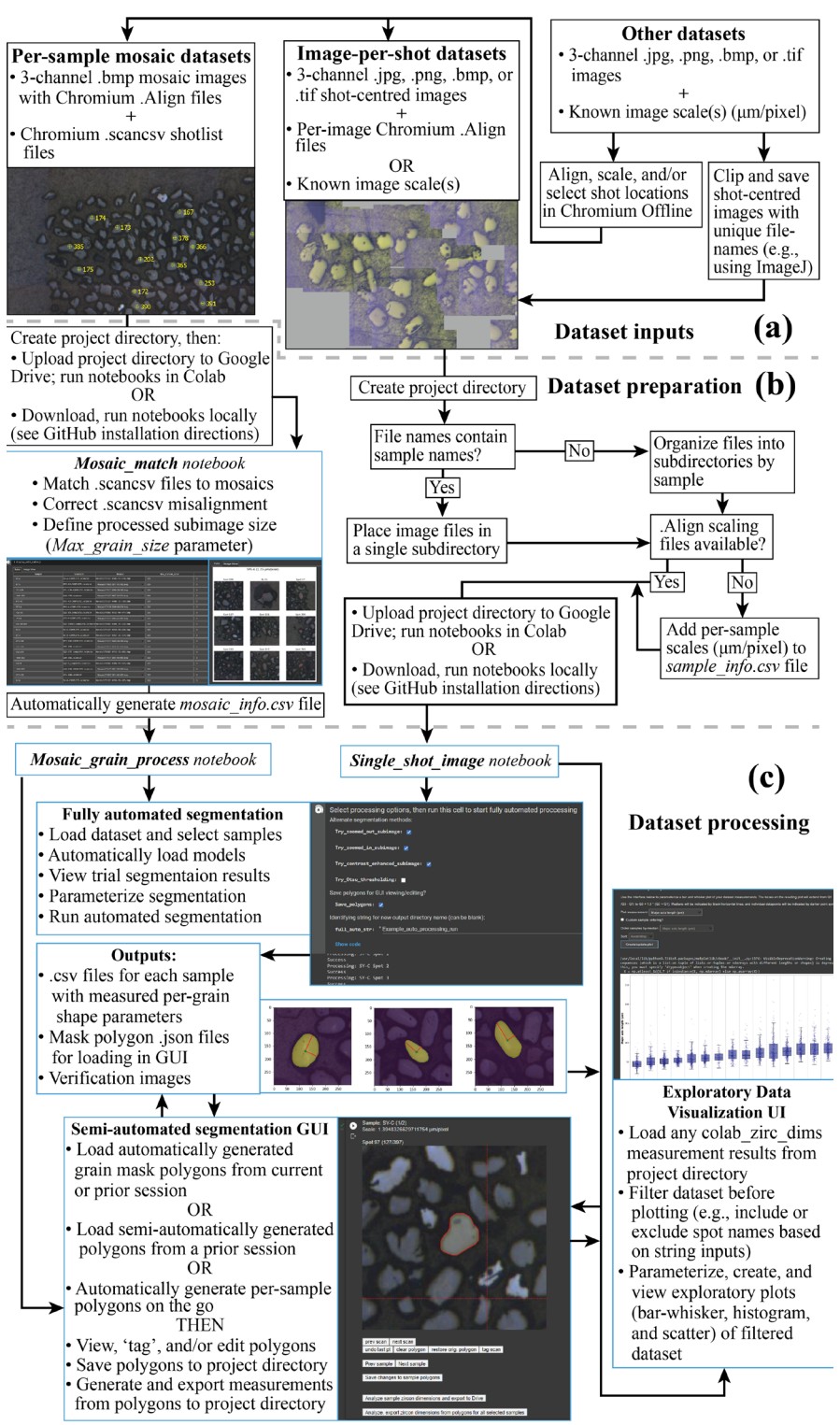

## 4.3 Notebooks

### 4.3.1 Dataset preparation tools

As we note in Sect 3.4, segmentation and measurement of grains in mosaic image datasets requires extraction of shot-specific sub-images from larger mosaics using shot locations in corresponding .scancsv shot metadata files. Information on which mosaic file in a project folder matches which .scancsv file must consequently be provided by users for processing. Because deep learning models struggle to identify and segment grains when they cannot see all grain boundaries (e.g., if sub-images are smaller than grains), sub-image extraction also requires a user-provided, mosaic-specific sub-image size parameter ("Max_grain_size") for accurate segmentations and measurements. Colab_zirc_dims processing notebooks read the aforementioned information from "mosaic_info" .csv files stored in project folders. Although these "mosaic_info" files can be created manually, they can also be generated quickly and easily using the "*Mosaic_Match*" colab_zirc_dims notebook (Fig. 2b) that we provide. The "*Mosaic_Match*" notebook implements code that automatically finds matches between shot lists and mosaics in a project folder and allows users to generate, modify, and export "mosaic_info" tables (Fig 2b). Users can view sample shot locations and sub-images using a "Display" function (Fig. 2b), thus allowing interactive mis-alignment correction, adjustment of sub-image sizes, and, in cases where multiple mosaics could potentially match a single .scancsv file, identification and selection of the correct mosaic from a dynamically populated dropdown menu. After exporting a "mosaic_info" .csv file, users can proceed to fully or semi-automated segmentation and measurement of their dataset (Figs. 2b, 2c).

### 4.3.2 Fully automated segmentation and measurement

We provide notebooks for automated and semi-automated processing of both mosaic image ("*Mosaic_grain_process*") and per-shot image ("*Single_shot_image_grain_process*") datasets. These notebooks are respectively currently set up to fully support processing of ALC and UCSB datasets but will likely work with datasets from other facilities sans-modification. The per-shot image notebook additionally supports loading and processing of any grain-centred reflected light grain images without Chromium scaling metadata, in which case users can provide custom per-sample scaling information in a .csv file or use a default scale of 1 µm/pixel. Researchers with datasets comprised of reflected light images that are not shot-centred and lack Chromium metadata can adapt (i.e., Fig 2a) their image datasets for use with colab_zirc_dims. This can be done either by using Chromium Offline (Teledyne Photon Machines, 2020) to generate scaling and/or shot placement metadata or by manually cropping shot-centred images from mosaics (e.g., using ImageJ's "multicrop" function; Schindelin et al., 2012). Such a workflow (Fig. 2a) will, however, bypass most of the automation in the colab_zirc_dims data loading process, and potential users are advised that collecting grain measurements using other existing software (i.e., AnalyZr; Scharf et al., 2022) will likely be less arduous.

Deep learning segmentation model weights are selected by users from a dropdown menu and downloaded to virtual or local machines from an Amazon Web Services S3 repository (provided by us) prior to model initialization and processing.

After weight file download and model initialization, users can select options for automated processing (Fig. 2c). These options include whether to attempt segmentation with various alternate methods (e.g., zooming out slightly, increasing image contrast before reapplying the model, or, as a last resort, using Otsu thresholding) if segmentation is initially unsuccessful, and whether to save polygons approximating model-produced masks for viewing or modification in the colab_zirc_dims graphical user interface (GUI; Fig 2c). During automated processing, per-grain dimensional analyses (Sect. 3.3) in per-sample .csv files are saved and exported to the user's project folder (Fig.2c) alongside verification mask image .png files (e.g., Figs. 1e, A1c).

### 4.3.3 Notebook-based GUI for semi-automated segmentation and measurement

We provide a simple, notebook-based GUI (Fig. 2c) extended from code in the Tensorflow Object Detection API (TensorFlow Developers, 2022) that allows users to view, modify, and save polygon-based grain segmentation masks. These polygon-masks can either be loaded from a previous automated or GUI-based processing session or generated on-the-fly on a per-sample basis. After viewing or re-segmenting part or all of a dataset, users can send their grain segmentations for measurement and export (Sect. 4.3.2); grain dimension exports from the GUI will include additional tags indicating whether each grain was segmented by a human or by a deep learning model.

### 4.3.4 Notebook-based exploratory data visualization interface

We do not provide any tools for assessing relationships between grain size or shape and age. Our processing notebooks do, however, include a simple interface that allows users to interactively load and filter (e.g., by scan name) colab_zirc_dims measurement data from their project folder before visualizing said data using parameterizable bar-whisker, histogram, and scatter plots (Fig. 2c).

### 5 Accuracy evaluations

We assessed the accuracy of our segmentation models by comparing a manually generated grain-dimension dataset (Leary et al., 2022) to automatically generated grain dimensions from the same samples measured using colab_zirc_dims. The test dataset from Leary et al. (2022) consists of samples collected from late Palaeozoic strata exposed across Arizona, USA. These samples were deposited in the same orogenic system—the Ancestral Rocky Mountains—as the Leary et al. (2020a) training dataset, and the grain ages and depositional environments are largely similar. The test dataset is unrelated to the training dataset from UCSB (see above). The full dataset was automatically processed using model M-ST-C and pure Otsu thresholding via the colab_zirc_dims "*Mosaic_Process*" notebook and the resulting automated *best long axis length* and *best short axis length* measurements were compared to the manual (measured with the Chromium "Measure" tool) per-grain axial measurements from the same dataset. For an n=301, sample-stratified random sub-sample of the Leary et al. (2022) dataset, colab_zirc_dims measurements of manual segmentation masks generated using the colab_zirc_dims semi-automated measurement GUI were also evaluated.

Table 2. Evaluation of error in colab_zirc_dims 'best axis' length measurements, with human measurements in the Leary et al. (2022) dataset used as 'ground truth'. For the full dataset (top), measurements produced by fully automated segmentation (using model M-ST-C) are compared against a baseline of Otsu thresholding. For the n=301 sample-stratified random subsample (bottom), measurements resulting from automated segmentation by model M-ST-C are compared to those resulting from new manual segmentations of the dataset using the colab_zirc_dims semi-automated processing GUI. Per-dataset best results on each metric are shown in bold type.

| Dataset | Model / method | n[a] | Failure rate[b] (%) | Average error[c] (µm) | | Average absolute error[d] (µm) | | Average error[e] (%) | | Average absolute error[f] (%) | | ≥ 20% absolute error rate[g] (%) | | Grain extent underestimate rate[h] (%) | Average segmentation time per image[i] (s) |
|---|---|---|---|---|---|---|---|---|---|---|---|---|---|---|---|
| | | | | Long axis | Short axis | Long axis | Short axis | Long axis | Short axis | Long axis | Short axis | Long axis | Short axis | | |
| Full[j] | M-ST-C | 5003 | **0.02** | **-2.3** | **-1.2** | **5.7** | **4.3** | **-2.06** | **-1.41** | **7.28** | **8.57** | **7.64** | **9.85** | **10.91** | 0.114 |
| | Otsu thresholding | 5003 | **0.02** | -6.1 | -5.7 | 10.1 | 7.8 | -7.41 | -10.94 | 13.03 | 15.82 | 18.19 | 25.78 | 27.00 | **0.011** |
| Random sub-sample[k] | M-ST-C | 301 | **0.0** | **-2.9** | **-1.2** | **6.4** | **4.5** | **-2.71** | **-1.58** | **7.95** | **9.09** | 8.64 | **9.97** | 10.30 | **0.137** |
| | Manual segmentation[l] | 301 | **0.0** | 3.6 | 3.9 | **6.4** | 4.8 | 5.84 | 8.73 | 8.62 | 10.65 | **8.64** | 14.29 | **1.66** | ~20 |

[a] Number of scan-images within dataset where a "central" grain mask could be identified with confidence ≥ 70%.

[b] $100 * \left(\frac{n_{total}-n}{n_{total}}\right)$

[c] $1/n \sum_{i=1}^{n}(axis_{measured})_i - (axis_{Leary\ et\ al.})_i$

[d] $1/n \sum_{i=1}^{n} \left|(axis_{measured})_i - (axis_{Leary\ et\ al.})_i\right|$

[e] $100 * \frac{1}{n}\sum_{i=1}^{n} \frac{(axis_{measured})_i - (axis_{Leary\ et\ al.})_i}{(axis_{Leary\ et\ al.})_i}$

[f] $100 * \frac{1}{n}\sum_{i=1}^{n} \left|\frac{(axis_{measured})_i - (axis_{Leary\ et\ al.})_i}{(axis_{Leary\ et\ al.})_i}\right|$

[g] $100 * \frac{number\ of\ grains\ with\ |\%\ error_{either\ axis}|\geq 20\%}{n}$

[h] $100 * \frac{(number\ of\ grains\ with\ error_{either\ axis}\leq -20\%)}{n}$

[i] Average time for model/method to successfully segment an image and return a measurable mask. Actual per-image processing times will be higher due to additional automated mask measurement and verification image saving time. Measured in Colab notebook with NVIDIA T4 GPU.

[j] The full Leary et al. (2022) dataset, with 5004 valid measurements.

[k] A sample-stratified random subsample of 301 measured grains from the Leary et al. (2022) dataset.

[l] By the first author, using the colab_zirc_dims semi-automated segmentation GUI in Google Colab.

**5.1 Machine error**

Otsu thresholding as implemented in colab_zirc_dims is a reasonably performant baseline segmentation method and apparently produces dimensionally accurate masks for the majority of grains in the Leary et al. (2022) dataset (Table 2). Our default model, however, significantly outperforms the baseline method of Otsu thresholding in every metric except for speed (Table 2). Given that segmentation time for M-ST-C is still a fraction of a second (Table 2) when run on a GPU-equipped computer,

deep-learning-based instance segmentation appears to be superior for producing high-quality segmentation masks from reflected light images. The Leary et al. (2022) image dataset is also mostly free of artefacts (e.g., Fig. A1), and we expect that the gulf in accuracy between the two methods would widen if evaluated on a lower-quality dataset.

**Figure 3. Plots displaying error distributions when comparing measurements produced by automated (M-ST-C) colab_zirc_dims**
**segmentation against manual measurements (i.e., Leary et al. 2022). (a) Automated (y-axis) versus manual (x-axis; Leary et al., 2022) measurement plots for long and short grain axes with linear regression lines plotted and gaussian KDE density shown via heatmap. Root mean squared error (RMSE) is shown at the bottom-right of each plot. (b) Histogram-KDE plots showing error distributions along long and short axes. Statistical information is shown at the bottom right of each plot.**

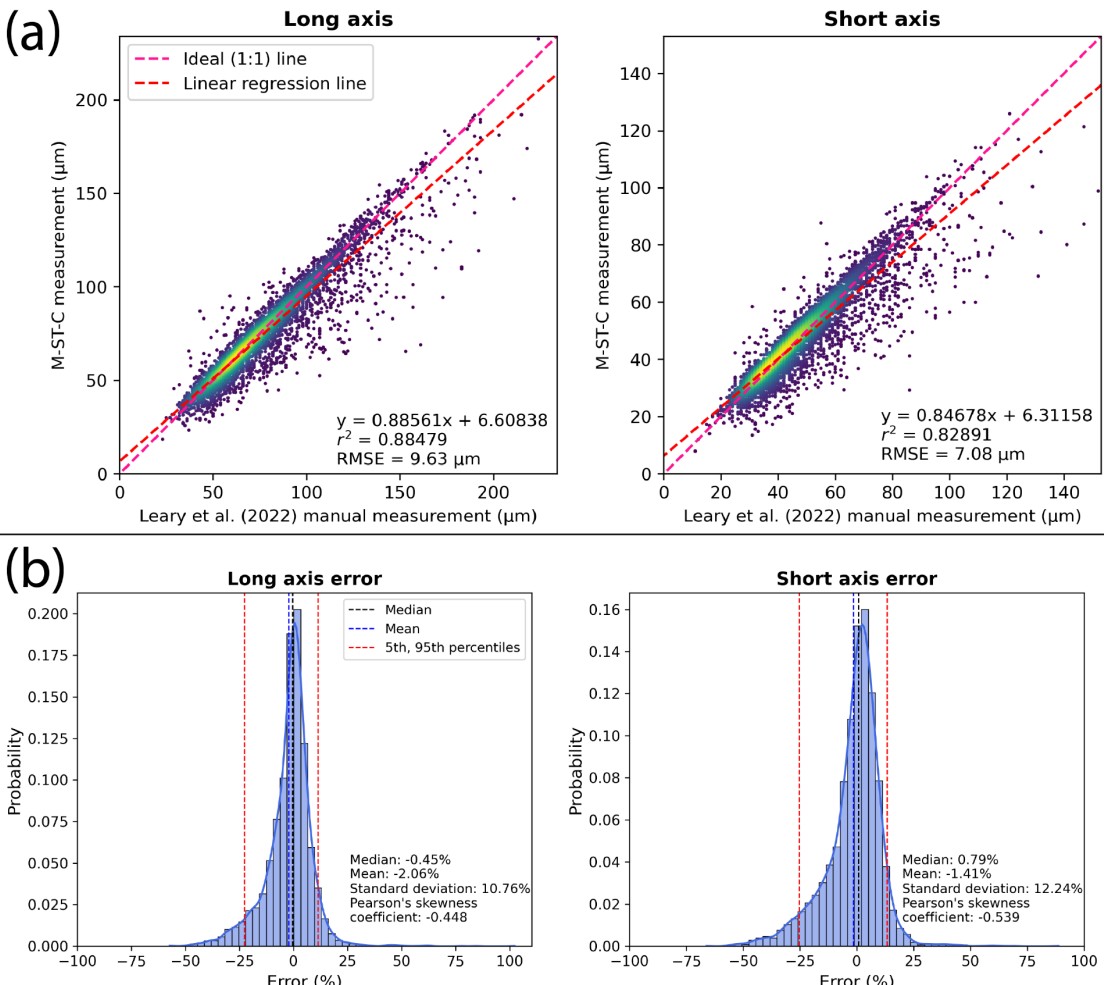

Per-grain automated (M-ST-C) measurements for the full Leary et al. (2022) dataset generally hew close to ground-
truth measurements but with a significant number of datapoints plotting well below the 1:1 measured versus ground truth (i.e., Leary et al., 2022) line (Fig. 3a). The apparent dominant cause of this negative skew (i.e., Equation 2; Fig. 3b) is under-segmentation of grains that are incompletely exposed at the surface of epoxy mounts but whose full grain areas are interpretable by humans from "shadows" visible in the (mostly) reflected light images (Fig. 4). We did not train our model to interpret

beyond clearly visible grain boundaries and it consequently fails to reproduce human measurements for these grains, but models might be able to do so without diminished accuracy on "normal" grains given training on a more interpretively segmented training dataset. Positive measurement errors are rare (Figs. 3A, 3B) but are probably mainly attributable to segmentation masks that merge different grains (Fig. 4). Failure to identify the correct "central grain" in images (Fig. 4) is likewise rare but may cause positive, negative, or negligible measurement error depending on the respective sizes of the target and mistakenly identified grains. Cases where no grain could be identified are exceedingly rare (Table 2, Fig. 4) and do not contribute directly to measurement error but, like all identified errors, necessitate manual re-segmentation of grains for production of accurate measurements.

**Equation 2:**

$$Pearson's\ skewness\ coefficient = \frac{3(mean - median)}{standard\ deviation}$$

**Figure 4. Examples of automated (M-ST-C) segmentation mask error modes with estimated occurrence rates, with axes scaled in μm and correct grain segmentations outlined in light blue. Rates for "grain boundary underestimate" and "no central grain found" errors are estimated from analysis of the entire Leary et al. (2022) dataset (i.e., Table 2). No "grain merging" or "wrong central grain" errors were identified in a manual review of the n = 301 sample of the full dataset (i.e., Table 2), and their occurrence rates are estimated from their non-appearance therein.**

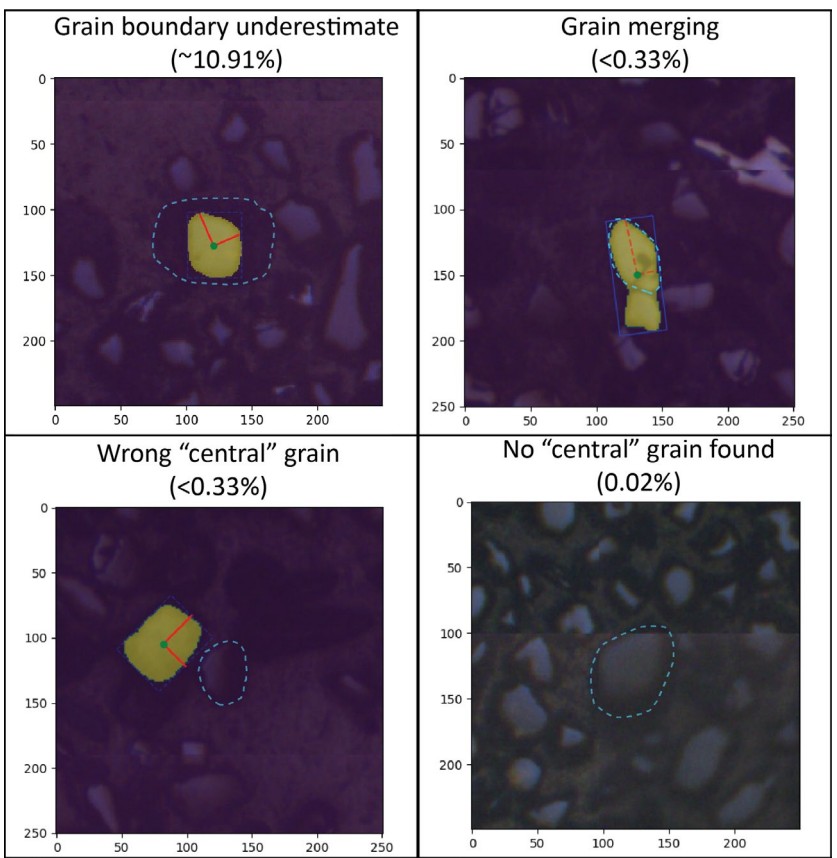

## 5.2 Human error

Automated measurement error metrics (e.g., Table 2) likely encompass some error that would be present even if grains were manually segmented, due to differential interpretations of grain areas between researchers. In the n = 301, randomly picked sample-stratified grain subsample from the Leary et al. (2022) dataset, we find that our default automated segmentation model (M-ST-C) achieves similar axial measurement absolute error metrics to the first author (M.S.) of this manuscript (Table 2). Though apparently mostly free of interpretive grain extent underestimates, the first author's measurements tend to be larger than dataset measurements (Table 2). Apparent over-interpretations of grain extents by the first author likely reflect different image display conditions (e.g., higher zoom and different contrast) during manual re-segmentation versus those present during collection of dataset measurements. Various features of colab_zirc_dims, namely automated segmentation of most grains and uniform image display conditions during manual segmentation of other grains, may enhance grain measurement dataset reproducibility in addition to collection speed.

## 5.3 Impact of grain exposure

We find that automated processing using colab_zirc_dims and our default model (M-ST-C) can approximately reproduce aggregate long and short grain axis length distributions for most samples in the Leary et al. (2022) mosaic image and measurement dataset (Fig. 5). Systemic negative errors along both grain axes are concentrated within four samples (1WM-302, 5PS-58, 2QZ-9, and 2QZ-272; Fig. 5). We found that grains in these samples were consistently underexposed above mount surfaces and that "grain extent underestimate" (Table 2; Fig. 4) segmentation errors were as a result common enough to negatively impact sample axis length distributions. Because these images are of sufficiently high quality that subsurface grain extents were interpretable by Leary et al. (2022), and because model M-ST-C generally only segments grain areas above resin surfaces, errors in these samples can also be used as a proxy for dimensional data loss from using reflected light versus transmitted light images to measure shapes of very poorly exposed grains in cases where reflected light images do not reveal any information about subsurface grain extents (Sect. 1; Leary et al., 2020a). In the worst-evaluated sample, 1WM-302 (n=180), M-ST-C produces axial measurements that underestimated manually measured grain axes by at least 20% 66.6% of the time, with average grain measurement errors of -18.0% and -22.0% along long and short axes, respectively. Treating these automatically generated axial measurements as ground truth data could result in significantly flawed analysis of relationships between grain size and age. Such shape parameter underestimates present only a minor (though potentially time-consuming) problem for colab_zirc_dims users with poorly exposed grains whose actual areas are still interpretable by humans (e.g., in the case of 1WM-302); erroneous segmentation masks can simply be corrected manually using the GUI. Users who observe that their mounted crystals are both very poorly exposed and invisible below the resin surface in their reflected light images, though, may consider re-imaging their samples using transmitted light and then measuring grains using a different program (e.g., AnalyZr) to avoid collecting flawed data. Researchers should consider excluding grain mounts that appear heavily over-

polished from their datasets, as accurate two-dimensional grain dimensions for these mounts will not be resolvable under any lighting conditions.

**Figure 5. Top: sample-by-sample boxplot comparison of human (Leary et al., 2022) and automated (M-ST-C) measurements along long and short grain axes. Below: additional scatter and bar-whisker plots showing relationships between human and automated grain long axis length measurements and U-Pb age, with samples binned by depositional period. Bottom: a KDE plot of detrital zircon U-Pb ages in the Leary et al. (2022) dataset. Boxplot boxes extend from Q1 to Q3, and whiskers extend from Q1 - 1.5 * (Q3 - Q1) to Q3 + 1.5 * (Q3 + Q1); sample medians are indicated by black horizontal lines within each box.**

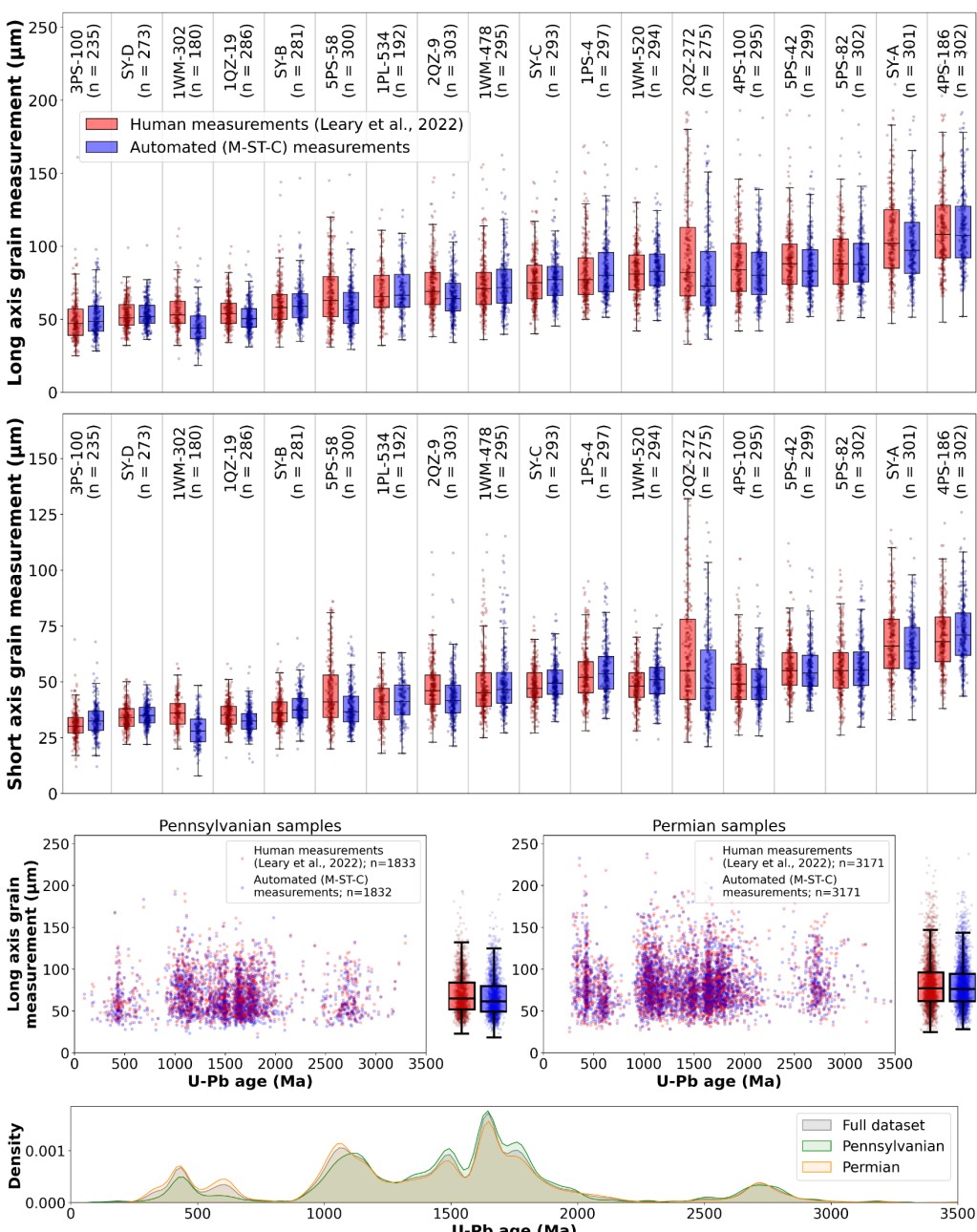

## 6 Viability of fully automated measurement

Due to low but significant segmentation error rates (Fig. 4) stemming almost entirely from poor grain exposure, we believe that manual segmentation verification and correction (i.e., semi-automated measurement) is necessary for production of publication-quality grain measurement datasets. Assuming time requirements of 35 minutes total to automatically generate segmentation masks, one second per grain to manually check masks, and 20 seconds to correct each mis-segmentation, and, conservatively (Fig. 4), that 15% of grains must be re-segmented via GUI, we estimate that it would take about six hours to semi-automatically collect zircon grain measurements for the full (n = 5,004) Leary et al. (2022) dataset using colab_zirc_dims.

We also believe, however, that fully automated measurement using colab_zirc_dims is a viable method for rapid approximation of grain dimensions in optimal samples (i.e., with well-exposed grains) as well as in larger datasets where the majority of samples have well-exposed grains. Meaningful relationships between grain dimensions and age appear to be resolvable solely based on fully automated measurement of such datasets. Leary et al. (2022) used zircon grain-dimension data to reinterpret the provenance and transport mechanism of 500-800 Ma zircons within the Pennsylvanian-Permian Ancestral Rocky Mountains system in southwest Laurentia. This reinterpretation was primarily based on the arrival of dominantly small (< 60 μm), 500-800 Ma zircons in that study area at the Pennsylvanian-Permian boundary. Leary et al. (2022) interpreted these grains as having been transported into the study area principally by wind and reinterpreted their provenance as Gondwanan (as opposed to Arctic and/or northern Appalachian as previously interpreted by Leary et al., 2020b). We find (Fig. 5) that this relationship is observable in fully automated (i.e., M-ST-C) measurement results from the dataset. Our hope is that the increased ability to explore such age-grain-dimension relationships and to generate large grain-dimension datasets from toolsets such as those presented here and by Scharf et al. (2022) will improve future provenance interpretations, specifically as they relate to grain transport processes (e.g. Lawrence et al., 2011; Ibañez-Mejia et al., 2018; Leary et al., 2020a; Cantine et al., 2021).

## 7 Limitations

Although our models (e.g., M-ST-C) evidentially generalize well to our test set, and we believe that they will most likely generalize well to other datasets, they are still untested on data from facilities not represented in their training dataset (i.e., besides ALC and UCSB). And, although they have been exposed to some relatively euhedral detrital zircon grains in the UCSB training images, our models are notably also untested on crystals derived from primary igneous and volcanic rocks. Some uncertainty remains in how well our models will work when applied to more diverse data by colab_zirc_dims users. We hope that any users who find that colab_zirc_dims struggles with their image data will share said data with us so that we can use it to expand on our training dataset and so improve our models' utility.

Measurements produced using colab_zirc_dims will persist all uncertainties that are innate to the methodology of measuring grain dimensions from reflected light images. Although most facilities aspire to polish their laser ablation zircon mounts to half the thickness of the zircons, it is possible that differences in sample preparation methods could produce

significant systematic interfacility or even intra-facility (i.e., between different analysts) biases in measurable two-dimensional grain dimensions; it remains somewhat unclear whether data derived through sample preparation and imaging at different facilities can be compared. Additionally, because there is some variability in quality of polish achieved at ALC in the test dataset (Leary et al., 2020a; see above discussion of samples 1WM-302, 5PS-58, 2QZ-9, and 2QZ-272), careful manual checking of polish quality will always be required in any dataset as described above. Ultimately, a study in which pre- (e.g. Finzel, 2017) and post-mount (Leary et al., 2020a; Scharf et al., 2022; current study) grain dimension measurements can be collected on the same samples, or one in which differential preparation methods are simulated (e.g., through slicing of three-dimensional micro-CT data, as applied to apatite by Cooperdock et al (2019)), will be the best way to quantify the bias introduced by polishing and/or by different facilities. However, such a test is well beyond the scope of the current study.

## 8 Future developments

The colab_zirc_dims package and Jupyter-style notebooks make it significantly faster and easier to augment an appropriate LA-ICP-MS dataset with grain measurements. We will continue to maintain and update colab_zirc_dims, and in the future hope to test and, if necessary, modify our code to extend full support to datasets from facilities beyond ALC and UCSB, possibly including those using targeting software other than Chromium. Although individual researchers are our intended userbase for colab_zirc_dims, we also believe that deep learning models hold great potential utility for LA-ICP-MS facilities. Such facilities are well-resourced to create large, customized training datasets and could implement trained models in a variety of applications including provision of per-spot grain measurements as a standard data product, fully automated spot picking, and possibly automated phase identification. Our training-validation dataset and pre-trained models (Sitar, 2022) may lower the barrier to entry for researchers and/or facilities hoping to apply machine- or deep-learning-based methods to similar problems.

## 9 Conclusions

We created a new, large dataset for instance segmentation of detrital zircon grain instances from reflected light images saved during LA-ICP-MS analysis. Using this dataset, we trained a suite of deep learning models and developed code that uses the models to rapidly extract per-grain dimensional measurements from LA-ICP-MS images collected at facilities using Chromium targeting software. We present this code as the colab_zirc_dims Python package, and we implement it in a collection of interactive Jupyter notebooks. These notebooks allow users to automatically or semi-automatically process datasets and can be run locally after installation of code dependencies or online in Google Colab with zero setup, hardware requirements, or installation.

The colab_zirc_dims deep-learning-based automated measurement algorithm approaches human measurement accuracy on a sample-by-sample basis and can be used to rapidly approximate grain size distributions for samples with well-

exposed zircon grains, without any human involvement. Our semi-automated segmentation workflow allows researchers to create manually reviewed and corrected grain size measurements for large-n datasets in under a day, although data collected through this process inherit all uncertainties related to the methodology of measuring mounted, polished grains in reflected

light images.

We believe that colab_zirc_dims makes it drastically easier to augment applicable LA-ICP-MS datasets with grain measurements, and hope that allowing more researchers to do so will expand our understanding of the relationships between zircon dimensions and age in varied environments. We also hope to extend full colab_zirc_dims support to datasets that do not currently work with its processing notebooks in the future and encourage users to share samples of such datasets with the first

author.

**Appendix A: Additional examples of segmentation results**

**Figure A1. Comparison between Otsu thresholding and CNN-based instance segmentation results in the presence of diverse grain morphologies and image artefacts, including anomalous bright spots (top row), heavily fractured grains (middle row), and tiling artefacts (bottom row). (a) Original grain-centred images clipped from ALC mosaics. (b) Segmentation masks produced via Otsu's**
**thresholding method (Otsu, 1979). (c) Instance segmentation results produced by a Mask RCNN model (M-ST-C) (at left) and resulting colab_zirc_dims verification image plots (at right).**

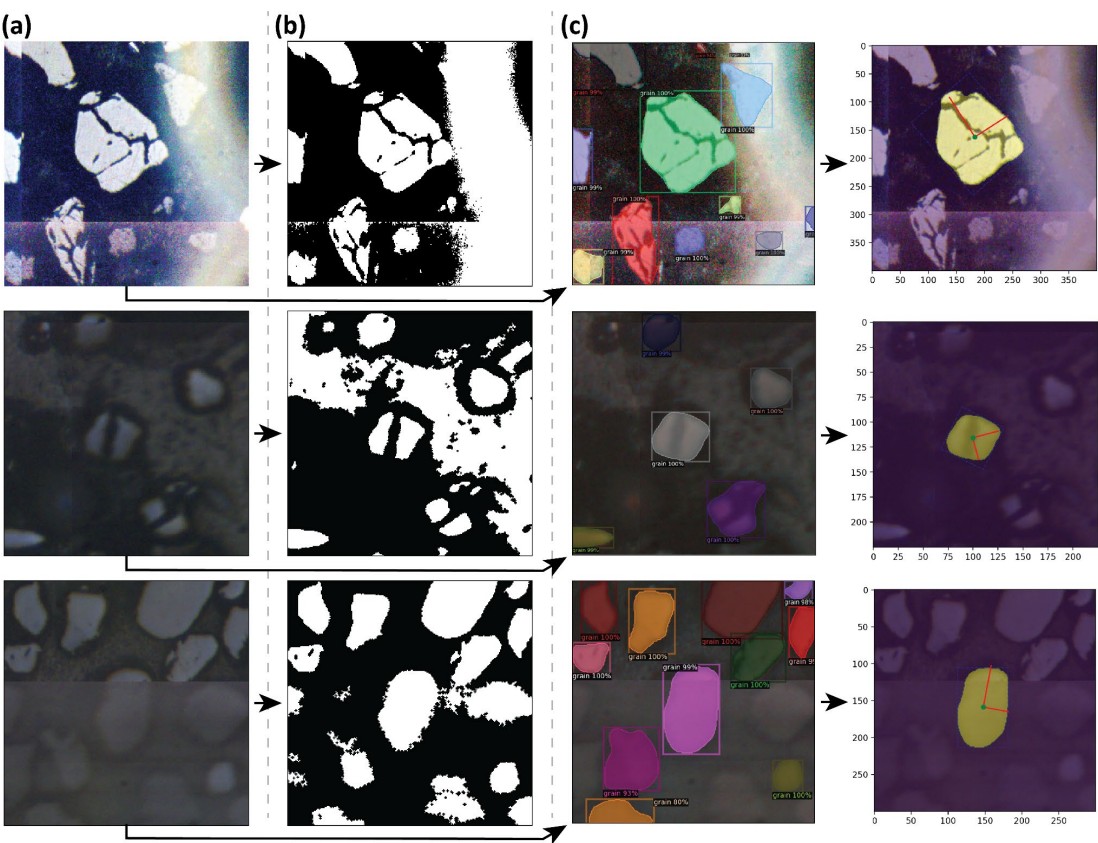

**Appendix B: Glossary of deep learning terminology**

**Table B1: A glossary of deep learning terminology used in this study.**

| Term | Explanation | Reference(s) |
|---|---|---|
| Weights | Training-optimizable parameters that are applied to data at various points within a neural network. | |
| Convolutional neural network (CNN) | A neural network wherein convolutional layers (roughly, these pass sliding filters over inputs) are used to abstract data. This allows processing of larger data (e.g., images) with fewer weight parameters. | |
| Backbone network | A module within a larger model that abstracts input data into an intermediate 'feature map' that is passed to other modules to produce the final model outputs. Larger model architectures are commonly referred to using the syntax "[model architecture name]-[backbone network name]". | |
| FPN | Feature Pyramid Network' -- a network that enhances feature maps via convolutional upsampling. Can be attached to a backbone network within a larger model to improve resolution of small objects. | Lin et al. (2016) |
| Mask-RCNN | A CNN-based model architecture developed by He et al. (2018). Internal modules use the feature map returned by a backbone network to propose regions which may contain objects. Later, independent modules fit bounding boxes to and create masks for each detected object. The most commonly used Mask-RCNN backbone is the ResNet network (He et al., 2015). | He et al. (2015, 2018) |
| Swin-T | Swin-'Tiny': the smallest variant of the 'Swin' model architecture (Liu et al., 2021), which is based on 'transformer' architecture (Vaswani et al., 2017). In transformer networks, inputs are respectively translated to and from a higher dimensional space by 'encoder' and 'decoder' modules.  These are impractical for direct application to images, as computational complexity scales exponentially with pixel count. The Swin architecture deals with this by splitting up image data using smaller, shifting windows. | Liu et al., (2021), Vaswani et al. (2017) |
| Centermask | A CNN-based model architecture developed by Lee and Park (2020). Similar to Mask-RCNN, except objects are detected and fit with bounding boxes by a single module, without an intermediate region proposal stage, prior to mask generation for each object. The standard backbone is VoVNet (Lee et al., 2019). | Lee et al. (2019), Lee and Park (2020) |

**Code availability**

The colab_zirc_dims source code, small example datasets, and links to pre-formatted template project folders and the latest versions of colab_zirc_dims Google Colab notebooks are available at the colab_zirc_dims GitHub page (Sitar, 2022): https://github.com/MCSitar/colab_zirc_dims. Additional code for reproducing error evaluations and figures presented in this manuscript using new or included automatically generated measurements is included in the supplementary data repository
(Sitar and Leary, 2022): https://doi.org/10.5281/zenodo.7434851.

**Data availability**

The full Leary et al. (2022) dataset of images and measurements that we used for model evaluation, our training dataset, and full measurement and evaluation data supporting the results presented in our manuscript can be found in the supplementary data repository (Sitar and Leary, 2022): https://doi.org/10.5281/zenodo.7434851.

**Author contribution**

M.S. wrote the first draft of the manuscript and both authors contributed to subsequent drafts. M.S. segmented the training dataset, trained the models, developed the code, and evaluated model-derived measurements. R. L. provided contextualized image and measurement datasets for model training and evaluation and feedback for improvement of the code and processing notebooks.

**Competing interests**

The authors declare that they have no conflict of interest.

**Acknowledgements**

We would like to thank Kurt Sundell for insights into imaging systems at the Arizona LaserChron Center. We would also like to thank Simon Nachtergaele, Taryn Scharf, and Nikki Seymour for their thoughtful reviews, which helped us to improve our manuscript considerably. Michael Sitar is additionally grateful to John Singleton for the leeway to finish this project. Previously unpublished UCSB training image data were collected in collaboration with Alaina Rosenthal-Guillot, with assistance from Andrew Kylander-Clark. Work by Michael Sitar was completed concurrently with work supported by grants from the National Science Foundation (award # 2115719) and USGS EDMAP Program (award # G21AC10493).

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
