# Peer review of "Technical Note: colab zirc dims: a Google-Colab-compatible toolset for automated and semi-automated measurement of mineral grains in LA-ICP-MS images using deep learning models"

_Geochronology, 2022_

## Author Response (AR1)

Dear Dr. Vermeesch,

We want to thank you for your constructive comments on our manuscript and for the several long extensions that you and the other Geochronology editors have granted us over the past several months. Your suggestions have been incorporated in a majorly revised manuscript and via several additional pages on the colab_zirc_dims GitHub page. Please see below for our responses (plain text; with occasional references to archived emails that we have exchanged) to each of your comments (italicized):

1. *I found the paper easy to read up to Section 3.2, when its complexity suddenly increases. Here the text contains a frightening number of acronyms and technical terms, including Mask RCNN, MS COCO, NMS, Detectron2, Swin-T, FPN, ResNet50, ResNet-101, "backbone network", Centermask etc. There are three problems with this complexity. First, the AI jargon won't make sense to the vast majority of GChron readers, who are not experts in this field. Second, given the rapidly changing landscape in AI technology, it won't be long before the specific tools used in colab_zircon_dims are superseded by more performant alternatives. Therefore, even AI experts may have trouble understanding the paper in the future. Third, whilst it is easy to update software to keep up with technical developments, the same is not true for academic papers. If you swap out some components in colab_zirc_dims, then the notebook will be 'out of sync' with the GChron paper. In order to make the paper more future proof, I suggest rewriting the text in a more generic form. Please explain the AI segmentation algorithm in general terms and dedicate fewer words to the specific implementation. Technical notes are meant to be short anyway, so much of the specific details could be moved to the online documentation of the Jupyter notebook. I appreciate that it is not possible or desirable to remove all jargon from the paper. It would be useful to add a table to the revised manuscript, listing the remaining definitions and acronyms.*

We have moved all discussion of differences between deep learning model architectures, training regimes, and their respective results when applied to our test datasets to a 'Model Library' subpage of the project GitHub page, which we refer to in our revised manuscript, with most remaining deep learning technical terms and associated references removed to an appendix table (B1). Granular information on our new training-validation dataset and on exported shape parameters are also now hosted on our project GitHub page. We sincerely appreciate your advice here; these edits should give us significant leeway to update models in the future without depreciating the manuscript. We hope that they also improve the interpretability of the text.

Regarding manuscript length: the net result of these edits is a three-page reduction in total manuscript length versus the manuscript version incorporating requested edits from the referees. The main body of our updated manuscript (i.e., ignoring the two appendices, which could be moved to our supplement if absolutely necessary) is approximately 5 pages shorter. We hope that this is closer, at least, to an appropriate length for a technical note.

2. *It is not clear how the apparent grain sizes measured by the AI algorithm on 2D images relate to the actual size distribution in 3D. The introduction mentions some published studies investigating age-size relationships in zircon U-Pb geochronology. Some of these studies (Lawrence et al., 2011) used sieves, whereas others (Cantine et al., 2021) used images. Are there any studies that have compared both approaches? I would imagine that their results can differ significantly. On a related note, the reviewers have already highlighted some issues caused by colab_zirc_dims's reliance on reflected light images. As pointed out in your manuscript, reflected light images of polishing surfaces tend to underestimate the actual grain sizes. Cross sectional areas also depend*

*on the polishing depth and on whether the grains are mounted as SIMSstyle epoxy pucks, or on glass slides (Figure 1). As a consequence, cross sectional grain sizes can only be used to compare samples prepared in the same lab and by the same analyst. I do not think that they can be used to compare samples from different labs or prepared by different analysts. This greatly reduces their usefulness. The revised manuscript needs to assess these limitations upfront.*

We basically agree with your concerns here, and especially appreciate the figure that you created to show how grain mounts themselves likely influence surface-level grain exposures. Also, to answer your question: we are not aware of any such study. In the latest version of our revised manuscript we include distinct "Impact of grain exposure" and "Limitations" sections that note potential uncertainties stemming from differential preparation methods and/or grain exposure and highlight the need for additional study. We also mention these uncertainties in the revised abstract and conclusion. In total, these changes will hopefully impart to readers the issues (both well-understood and unresolved) inherent to our approach to grain dimension measurement.

3. *According to Section 5.2 of the paper, it would take an estimated six hours to determine the grain size distribution of the Leary dataset. This is impractical. Unless the grain size measurements are truly effortless, I'm afraid that the AI approach won't get much use. I understand that the Jupyter notebook is a proof-of-concept product. What would need to be done to make it faster or easier and use?*

As noted in our prior email exchange, most of the time estimated here would be spent manually reviewing (~1.4 hours) and correcting (~4.2 hours) grain segmentations. The best way to make this process 'effortless', then, would be to train a model that makes very few mistakes when segmenting grains. I (the first author) sought to do this by creating a new, much larger grain image-annotation dataset (1558 images with 16464 grains annotated) and retraining models using it. This succeeded to some degree: probably as a direct result of the conservative annotation strategy used on the training-validation dataset, the most performant re-trained model still fails to interpret larger subsurface grain extents where suggested by subtle "shadows" surrounding some poorly exposed grains in the test dataset (i.e., from Leary et al., 2022). It is, however, practically free ($\leq 0.33\%$) from other segmentation error types. The estimated total time required to semi-automatically measure the ~5000-grain Leary et al. (2022) dataset using colab_zirc_dims is still, unfortunately, around 6 hours, largely dependent on surface-level grain exposure in images. For samples within the dataset with uniformly well-exposed grains, semi-automated processing is much quicker (~15 minutes/600 grains; as shown in the v1.0.10 video tutorial).

Correcting the under-segmentation errors within the confines of the existing colab_zirc_dims framework would require manually reannotating large portions (i.e., all grains with "shadows") of the new training dataset. Such is the double-edged sword of deep learning algorithms. We might do this in the future but elected not to at present; you might agree that post-preprint development of our new dataset and models has taken too long already. Our deep-learning-based models' performance when compared to algorithms that have been previously applied to this problem will hopefully still serve as an adequate proof of utility, persistent under-segmentation errors in some ALC images non-withstanding. As for the (admittedly, long) ~6 hour time estimate for semi-automated large-n dataset measurement: we agree that this probably discourages widespread adoption of colab_zirc_dims for grain measurement. We are, however, still cautiously hopeful that researchers who have already committed to the many hours of often-tedious work required to produce a large-n detrital zircon age dataset (e.g., ~60 hours of LA-ICP-MS time in the case of Leary et al. (2022)) will not be overly averse to spending several more measuring their grains.

4. *Section 3 lists the advantages and disadvantages of Google Colab. It does not mention two problems. First, Colab requires an internet connection, yet many lab computers are not connected to the web due for security reasons (automatic OS updates are disabled on most lab computers). Second, Google products are not accessible from China, which is a huge 'geochronological market'.*

We have addressed these issues by updating the code such that, given some hardware-specific additional installation steps (detailed on the project GitHub page), the code and notebooks can be run on local machines. Said instructions also detail an approach that *should* allow code and/or notebook execution on machines that are not connected to the internet, though I (first author) have not yet been able to properly test this on an out-of-date lab computer.

5. *In your response to Dr. Nachtergaele, you chose not to follow his suggestion to add a plot of grain size vs. U-Pb age to your paper, because such a plot is already scheduled to appear in an upcoming JSR paper by Leary et al. I would urge you to reconsider this decision, for two reasons. First, the Leary et al. study only presents the manual measurements, and not your automated results. Second, the paper will be stuck behind a paywall, so not all GChron readers will be able to check this useful figure. I suggest that you replace Figure 8 with a scatter plot of grain size vs. U-Pb age for a representative sample, with a box plot and KDE shown along the y- and x-axis, respectively.*

As noted in my prior email, this is now addressed with a compilation figure (Figure 5; expanded from Figure 8 of the pre-print manuscript) which illustrates the apparent reproducibility of grain age-shape relationships identified by Leary et al. (2022) using fully-automated measurement data. These relationships are explained in some detail within a new "Viability of fully automated measurement" section.

Thank you again for your time and constructive evaluation of our manuscript. We look forward to receiving your feedback on the revised copy!

Sincerely,
Michael C. Sitar and Ryan J. Leary

**Reference:**

Leary, R. J., Smith, M. E., and Umhoefer, P.: Mixed eolian–longshore sediment transport in the late Paleozoic Arizona shelf and Pedregosa basin, U.S.A.: A case study in grain-size analysis of detrital-zircon datasets, Journal of Sedimentary Research, 92, 676–694, https://doi.org/10.2110/jsr.2021.101, 2022.

---

## Referee Report (RR1)

Review of *Technical Note: colab_zirc_dims: a Google-Colab-compatible toolset for automated and semi-automated measurement of mineral grains in LA-ICP-MS images using deep learning models*
Author(s): Michael C. Sitar and Ryan J. Leary

In this manuscript, Sitar and Leary present a new set of tools to automate the measurement of zircon axes in polished laser ablation mounts. These freely-available, interactive tools can be locally installed or accessibly run through Google Colab. The authors detail the use of the tools as well as both the deep learning concepts behind the code and limitations users should consider while validating the measurements produced by these tools. First, I would like to state that I do not have expertise in deep learning and cannot speak to the choice of one thresholding method over another, grain-segmenting algorithms, or other technical choices. I am approaching this review as a zircon geochronologist with extensive Chromium experience interested in the user experience and feasibility of integrating the toolset into my workflow.

The authors clearly establish the need to consider grain measurements in detrital studies, and point out a gap in data that could be effectively and efficiently filled with their new toolset. The authors may want to include the 3D grain measurement efforts using MicroCT methods (e.g., Cooperdock et al., 2022; Cooperdock et al., 2019) - these have been applied in the low-temperature thermochronology realm rather than the laser ablation realm, but show the importance of grain size measurements across geochronometric methods. Indeed, manually measuring grains or the application of MicroCT is a time-consuming process, especially as large-n datasets proliferate, and reducing barriers to data collection will facilitate the inclusion of additional dimensions of data. The issue of sectioning bias remains –the authors acknowledge this bias, and are clear about the limitations of this code and need for users to evaluate the data.

The authors make a compelling case for the use of deep learning based techniques to bridge this gap. They acknowledge existing methods of identifying individual grains (ZirconSpotFinder and AnalyZr) and point out how their toolset differs (rapid automated measurements with a reduced need for many hours of user involvement; accessibility through either Jupyter or Google Colab).

I appreciate that the authors utilized training datasets from multiple labs and differing image collection approaches (per-sample mosaic versus image-per-shot). When it comes to the detailed discussion of segmentation techniques and deep learning models, I do not have the background necessary to follow the architectures/backbones or choices in model parameter optimization. I defer to other scientists with expertise in these fields, as the discussion of these matters is described with impenetrable levels of jargon (especially section 3.3). I see a previous reviewer ran into this same situation – the authors have added a glossary at the end of the manuscript, which helps, but does not fully alleviate the difficulty. The point raised in the previous review still stands.

The authors do step through what the toolset is actually doing (section 3.4 and 4, and the inclusion of Figure 2 is very helpful. I chose to run through the Google Colab files and test them for ease of use. I commend the authors on the detailed commentary within the code files and video tutorial! I also really like how they have linked the different notebooks together at the end of each "chapter". Limitations and warnings about how to produce publication-quality data are included at the relevant points in the code. I did miss the point about playground mode, so in an effort to keep users from inadvertently editing things they should not edit, is it possible to add a cell in the notebook that automatically kicks them over to playground mode? The top of my notebook did read "cannot save changes" as in the tutorial, so maybe this has already been implemented. The notebooks were easy to follow, and are fully suitable for someone with limited programming experience (i.e., students, etc). I tested them on some of my own images and once I structured the project directory appropriately, the code ran without an issue.

Cooperdock et al., 2022: https://gchron.copernicus.org/articles/4/501/2022/
Cooperdock et al., 2019: https://gchron.copernicus.org/articles/1/17/2019/

---

## Editor Decision (ED1)

Prof. Pieter Vermeesch
University College London
+44 (0)20 3108 6369
p.vermeesch@ucl.ac.uk

8 August 2022

To: Michael C. Sitar, Colorado State University

Dear Mr. Sitar,

Thank you for submitting your Technical Note entitled "colab_zirc_dims: a Google-Colab-based Toolset for Automated and Semi-automated Measurement of Mineral Grains in LA-ICP-MS Images Using Deep Learning Models" to *Geochronology*. Having considered your paper, the two reviews and your response to the reviews, I have decided that your manuscript is suitable for publication in *Geochronology* after revisions. In addition to the reviewer comments, I would like to add a few thoughts of my own.

1. I found the paper easy to read up to Section 3.2, when its complexity suddenly increases. Here the text contains a frightening number of acronyms and technical terms, including Mask RCNN, MS COCO, NMS, Detectron2, Swin-T, FPN, ResNet-50, ResNet-101, "backbone network", Centermask etc. There are three problems with this complexity. First, the AI jargon won't make sense to the vast majority of GChron readers, who are not experts in this field. Second, given the rapidly changing landscape in AI technology, it won't be long before the specific tools used in `colab_zircon_dims` are superseded by more performant alternatives. Therefore, even AI experts may have trouble understanding the paper in the future. Third, whilst it is easy to update software to keep up with technical developments, the same is not true for academic papers. If you swap out some components in `colab_zirc_dims`, then the notebook will be 'out of sync' with the *GChron* paper. In order to make the paper more future proof, I suggest rewriting the text in a more generic form. Please explain the AI segmentation algorithm in general terms and dedicate fewer words to the specific implementation. Technical notes are meant to be short anyway, so much of the specific details could be moved to the online documentation of the Jupyter notebook. I appreciate that it is not possible or desirable to remove all jargon from the paper. It would be useful to add a table to the revised manuscript, listing the remaining definitions and acronyms.

2. It is not clear how the apparent grain sizes measured by the AI algorithm on 2D images relate to the actual size distribution in 3D. The introduction mentions some published studies investigating age-size relationships in zircon U-Pb geochronology. Some of these studies (Lawrence et al., 2011) used sieves, whereas others (Cantine et al., 2021) used images. Are there any studies that have compared both approaches? I would imagine that their results can differ significantly. On a related note, the reviewers have already highlighted some issues caused by `colab_zirc_dims`'s reliance on reflected light images. As pointed out in your manuscript, reflected light images of polishing surfaces tend to underestimate the actual grain sizes. Cross sectional areas also depend on the polishing depth and on whether the grains are mounted as SIMS-style epoxy pucks, or on glass slides (Figure 1). As a consequence, cross sectional grain sizes can only be used to compare samples prepared in the same lab and by the same analyst. I do not think that they can be used to compare samples from different labs or prepared by different analysts. This greatly reduces their usefulness. The revised manuscript needs to assess these limitations upfront.

Prof. Pieter Vermeesch
University College London
+44 (0)20 3108 6369
p.vermeesch@ucl.ac.uk

[Figure]

[Figure]

*Figure 1: Some laboratories mount zircons in epoxy pucks (left), whereas others use glass slides (right). In both cases, the apparent grain size in cross section depends on the depth of the polishing surface (dashed line). Grain size variation leads to variable degrees of over- and under-polishing. This diminishes the ability to compare grain size distributions between analysts, and between laboratories.*

3. According to Section 5.2 of the paper, it would take an estimated six hours to determine the grain size distribution of the Leary dataset. This is impractical. Unless the grain size measurements are truly effortless, I'm afraid that the AI approach won't get much use. I understand that the Jupyter notebook is a proof-of-concept product. What would need to be done to make it faster or easier and use?

4. Section 3 lists the advantages and disadvantages of Google Colab. It does not mention two problems. First, Colab requires an internet connection, yet many lab computers are not connected to the web due for security reasons (automatic OS updates are disabled on most lab computers). Second, Google products are not accessible from China, which is a huge 'geochronological market'.

5. In your response to Dr. Nachtergaele, you chose not to follow his suggestion to add a plot of grain size vs. U-Pb age to your paper, because such a plot is already scheduled to appear in an upcoming *JSR* paper by Leary et al. I would urge you to reconsider this decision, for two reasons. First, the Leary et al. study only presents the manual measurements, and not your automated results. Second, the paper will be stuck behind a paywall, so not all *GChron* readers will be able to check this useful figure. I suggest that you replace Figure 8 with a scatter plot of grain size vs. U-Pb age for a representative sample, with a box plot and KDE shown along the y- and x-axis, respectively.

*Geochronology* normally gives authors four weeks to complete the revision. I would be happy to extend this if you need more time to address my first comment. Please do not hesitate to contact me if you have any questions.

Sincerely yours,

Pieter Vermeesch
Associate Editor
*Geochronology*

---

## Author Response (AR2)

Dear Dr. Vermeesch,

We would like to extend our deepest thanks for your assistance and feedback throughout the lengthy review process for this manuscript. We likewise are grateful for the opportunity to have our work published in Geochronology and look forward to contributing to the field with our research and code.

As noted by both you and the final reviewer, some of the sentences in the previously submitted version of our manuscript are quite long. We have shortened and/or split these up as possible in our finalized manuscript. Additionally, we have added one of the two references suggested by the final reviewer and cleaned up some other citations and typos besides. We hope that these changes improve our manuscript's the clarity and readability.

Sincerely,

Michael Sitar and Ryan Leary